# Antimicrobial Activity of Frankincense (*Boswellia sacra*) Oil and Smoke against Pathogenic and Airborne Microbes

**DOI:** 10.3390/foods12183442

**Published:** 2023-09-15

**Authors:** Zahra S. Al-Kharousi, Ann S. Mothershaw, Basil Nzeako

**Affiliations:** 1Department of Food Science and Nutrition, College of Agricultural and Marine Sciences, Sultan Qaboos University, P.O. Box 34, Al-Khod 123, Oman; asmothershaw@hotmail.com; 2Department of Microbiology and Immunology, College of Medicine, Sultan Qaboos University, P.O. Box 35, Al-Khod 123, Oman; basil@squ.edu.om

**Keywords:** antimicrobial activity, *Boswellia sacra*, frankincense, particle size, pathogens, medicinal plants, smoke

## Abstract

As they continuously evolve, plants will remain a renewable source for antimicrobial compounds. Omani frankincense is produced by *B. sacra* trees and is graded into Hojari, Nejdi, Shazri or Sha’bi. Air can be a source for pathogenic or food spoilage microbes; thus, inactivating airborne microbes is necessary in environments such as food and animal production areas. This study investigated the antimicrobial activity and the chemistry of steam-distilled oils of Hojari and Sha’bi grades. It also analyzed the antimicrobial activity of frankincense smoke and the size of its solid particles. Chemical analysis was performed using gas chromatography mass spectrometry (GC-MS). The antimicrobial activity of the oils against *Staphylococcus aureus* (NCTC 6571), *Bacillus* spp., *Escherichia coli* (NCTC 10418), *Pseudomonas aeruginosa* (NCTC 10662), *Saccharomyces cerevisiae*, *Candida albicans*, *Aspergillus flavus*, *Aspergillus ochraceus*, *Aspergillus niger*, *Penicillium citrinum*, *Alternaria alternata* and *Fusarium solani* was determined using well diffusion and micro-well dilution methods. A microscopic technique was used to determine the size of frankincense smoke solid particles. Microbes were exposed to frankincense smoke to test their susceptibility to the smoke. Hojari and Sha’bi oils were similar in composition and contained monoterpenes and sesquiterpenes. The Hojari and the Sha’bi oils possessed broad spectrum antimicrobial activity. The largest growth inhibition zones were obtained with *S. cerevisiae* and *F. solani*. An MIC of 1.56% (*v*/*v*) was found with *E. coli*, *S. cerevisiae* and *F. solani*. Frankincense smoke contained fine irregular solid particles with a diameter range of 0.8–2287.4 µm, and thus may pose a health risk to susceptible individuals. The smoke had potent antimicrobial activity against *S. aureus*, *E. coli*, and airborne bacteria, yeast and mold, with a maximum inhibition of 100%. It was concluded that Hojari and Sha’bi frankincense oils and smoke had significant antimicrobial activity that can be exploited in controlling human, animal and plant pathogenic microbes.

## 1. Introduction

Plants have been the building blocks for traditional and modern medicinal systems [1] and will continue to be a renewable source for new drugs in the future [1,2,3]. A medicinal plant contains mixtures of various chemical compounds that may act synergistically, additively, or individually. Antimicrobial compounds that are derived from plant extracts can be beneficial in combating resistant strains as they can exhibit a different mechanism of action on pathogens than conventional drugs [2,4,5]. Essential oils were found to be active against bacteria, viruses, fungi, protozoa, worms and insects [6]. Essential oils are secondary metabolites that are produced by many aromatic or medicinal plants in small quantities. They are composed of mixtures of volatile compounds [7], and their synthesis can be constitutive or inducible from inactive precursors in response to stress [8]. They can be obtained by hydro- or steam distillation [6], chemical extraction or expression methods. Adding fresh or crushed herbs and spices directly to food can enable people to obtain essential oils [9].

Frankincense or olibanum is one of the oldest fragrant and medicinal plants known throughout the world [10,11]. It is a natural oleo-gum resin obtained from wounds made in the bark of *Boswellia* trees [12]. The genus *Boswellia* of the *Burseraceae* family includes about 23 species of small trees that grow mainly in Arabia, on the eastern coast of Africa, and in India [13,14,15]. The oleo-gum resin of frankincense is composed of a mixture of essential oils (mono- and sesquiterpenes), alcohol-soluble resins (diterpenes and triterpenes) and water-soluble gums (polysaccharides) [14,16,17]. *B. sacra* was the principal species used in classical times [17]. *B. sacra* trees grow in the southwest of Oman, in the Mahra and Hadramawt regions of Yemen, and in Somalia [18].

The oleo-gum resin of frankincense is burnt for its fragrance. Its essential oil is used in aromatherapy, and different formulae of frankincense are prepared and used by different cultures to treat numerous diseases [19]. Indians have reported the beneficial capabilities of frankincense since 5000 years ago when the powder and extracts of *B. serrata* were used as anti-inflammatory agents [12]. Frankincense has also been used to treat back pain, chronic cough, leukemia, and liver and colon cancers [16,19]. The methanol extract of *B. papyrifera* possessed good activity against methicillin-resistant *Staphylococcus aureus* (MRSA) [5], with a minimum inhibitory concentration of 62.5–500 µg/mL. The essential oil and extracts of frankincense are widely used as fixatives in soaps, lotions, perfumes and creams [20]. In Oman, *B. sacra* trees are indigenous to the Dhofar region, where they grow in four ecological zones, Hojar, Nejd, Shazr and Sha’b, and thus, there are four main grades: Hojari, Nejdi, Shazri and Sha’bi [19]. The tree is a very important wild resource and a symbol for the culture and history of Oman and can withstand the persistent extreme conditions of drought [18]. The estimated annual production of Omani frankincense is 80–100 tons gathered from about 500,000 trees [10].

Incense is burnt to create a pleasant smell as well as for sanitation purposes as it is believed to repel mosquitoes, flies, and bees. In Ethiopia, water and milk vessels are disinfected with frankincense smoke [16]. In Oman and other Gulf countries, incense that is burnt is called “bakhour”, and its ingredients include frankincense, flowers, herbs, essential oils, and aromatic woods such as sandalwood [21]. Incense burning is a long, slow, incomplete combustion process that results in the generation of a continuous smoke stream with the emission of PM10 and PM2.5 (particulate matter less than 10 and 2.5 μm in aerodynamic diameter), carbon monoxide (CO) and volatile organic compounds (VOCs) [22]. These chemicals easily accumulate indoors, especially when there is inadequate ventilation [21]. Indeed, epidemiological studies have reported an association between air particulate matter and several acute health effects, including mortality, respiratory symptoms and lung dysfunction [23].

The hypothesis of this study was that frankincense oil and smoke possess antimicrobial properties. The objectives of this investigation were to test and compare the ability of two grades (Hojari and Sha’bi) of Omani frankincense oil to inactivate foodborne microorganisms, to determine their minimum inhibitory concentration (MIC) and minimum bactericidal/fungicidal concentrations (MBCs/MFCs), to test if fumigation with frankincense has any effect on the survival of airborne microbes, to assess the size of frankincense smoke particles, and to analyze the chemical composition of the essential oils as an aid to account for any antimicrobial activity of the essential oils and to identify any variations in the chemical composition of different frankincense samples. Thus, this study screened the antimicrobial activity of frankincense oil using the well diffusion method and determined its MICs, MBCs, and MFCs. This is in contrast to some previous studies that reported the antimicrobial activity of the volatile constituents of essential oils [24] and the antimicrobial activity of frankincense hydro-distilled frankincense essential oil by the determination of the MIC only [13]. This study also allowed better a characterization of the antimicrobial activity of frankincense oil by using the broth micro-dilution method coupled with optical density readings. Moreover, to our knowledge, this is the first study to have developed a method to test the antimicrobial activity of frankincense smoke against airborne microbes. This study analyzed the chemical profile of the steamed-distilled frankincense oil in comparison to the previous studies that focused more on the determination of the phytochemical profile of hydro-distilled frankincense oil and showed the presence of various monoterpenes and sesquiterpene [25].

## 2. Materials and Methods

### 2.1. Collection of Frankincense Samples

Two grades of Omani frankincense (Hojari and Sha’bi, Appendix A) were purchased from a trusted Dhofari frankincense trader. Hojari and Sha’bi grades were selected to represent the top and the lowest grades of Omani frankincense, respectively. It was noted that these two grades are produced by frankincense trees growing in different geographical locations [19]. The trees that produce Sha’bi frankincense are more affected by monsoon rains. Four samples of Hojari grade were obtained, three of which were sub-graded as Hojari White (HW1, HW2 and HW3) and one was sub-graded as Hojari Green (HG). Five samples of Sha’bi were obtained (S1, S2, S3, S4 and S5). Sorting into different samples was performed by the trader, who noted that the HG sample was collected from frankincense trees growing in Wadi Habjar (Wadi Hojar); the HW1, HW2 and HW3 samples were collected from Wadi Andor, and the Sha’bi samples were gathered from Wadi Aful and Al-Mughsail. All samples were stored in the dark in their original plastic bags at room temperature until analyzed.

### 2.2. Extraction of Frankincense Essential Oil by Steam Distillation

Essential oils were extracted using the steam distillation method [26]. A Clevenger-type apparatus was used, which was composed of a distillation vessel, a cooling system (Cryo-bath, Chrompack) and a condenser. The condenser was equipped with a glass connector that allowed the hydrosol (frankincense water) to return to the distillation vessel. About 150 g of each sample was weighed using a top-loading balance (Precisa XB 620C, Dietikon, Switzerland) and then distilled for five hours. After condensation, the frankincense oil was decanted from the hydrosol. Distillation was performed for all nine samples separately and was completed in two weeks starting with S2 followed by S3, HW1, S5, HW2, S4, HG, S1, and HW3. The essential oils were collected in pre-weighed amber glass vials, closed tightly, and kept in the dark at 4 °C to prevent any alteration in their composition prior to analysis [9].

The weight of the collected oils was measured, and the percent yield (*w*/*w*) of essential oils was calculated [24] for each sample using the following equation:(1)$$\%F_{E}=\frac{W_{E}}{W_{F}}\times \;100$$
where % *F_E_* = the percent yield of frankincense essential oil (*w*/*w*), W_E_ = the weight of essential oil, and W_F_ = the original weight of frankincense oleo-gum resin.

### 2.3. Chemical Analysis of Essential Oil of Frankincense

Essential oils were chemically analyzed [25] within two months of their production date. Ten microliter of each essential oil sample was added into a clear screw-top GC-vial (6.0 mm diameter, silicone septa, 2 mL volume), and then 1.5 mL of dichloromethane was added to each vial. A blank containing only dichloromethane was included to confirm its purity. Essential oils were analyzed using a gas chromatography apparatus equipped with a quadruple mass spectrometer (GC-MS) (Shimadzu, GC-MS-QP/5050A). A nonpolar capillary column (DB-5MS, thickness; 0.25 μm, length; 30 m, internal diameter; 0.25 mm) was used to separate the components of the essential oils using a splitless mode. The temperature was kept at 51 °C for 1 min and then raised to 211 °C at a rate of 3 °C min^−1^. The injector temperature was 275 °C, and the interface temperature was 300 °C. Helium was used as the carrier gas. Chemical compounds were identified by matching their fragmentation patterns in the resultant mass spectra with those stored in the computer’s mass spectral database.

### 2.4. Antimicrobial Activity of Essential Oil of Frankincense

#### 2.4.1. Microorganisms

The test microorganisms were *S. aureus* (NCTC 6571), *Bacillus* spp., *Escherichia coli* (NCTC 10418), *Pseudomonas aeruginosa* (NCTC 10662), *Saccharomyces cerevisiae*, *Candida albicans*, *Aspergillus flavus*, *A. ochraceus*, *A. niger*, *Penicillium citrinum*, *Alternaria alternata* and *Fusarium solani*. *Bacteria* and *C. albicans* were obtained from the Department of Microbiology and Immunology, College of Medicine and Health Sciences, Sultan Qaboos University (SQU). *Bacillus* spp. was a wild-type strain, *C. albicans* was a clinical isolate, and *S. cerevisiae* was a commercial strain. *A. alternata* and *F. solani* were wild-type plant pathogens obtained from the Plant Sciences Department, College of Agricultural and Marine Sciences, SQU, while the other four molds were wild types obtained from the Biology Department, College of Science, SQU. Bacteria were maintained on nutrient agar (NA) slants at 4 °C, and yeasts and molds were maintained on potato dextrose agar (PDA) slants at 4 °C. The same bacterial and fungal strains were used throughout the study.

#### 2.4.2. Screening Antibacterial and Antifungal Activity of Essential Oil of Frankincense

Screening antimicrobial activity of oils was performed as described previously with some modifications [27]. Bacteria and yeasts were subcultured from slants onto NA and PDA plates, respectively. They were all incubated aerobically at 37 °C for 18–24 h (bacteria) and for 48 h (yeasts). The molds were subcultured onto PDA plates and incubated at 25 °C for 3–7 days until sporulation occurred. The agar well diffusion method was used to screen the antimicrobial activity of the essential oils against bacteria, yeasts and molds. Bacteria were subcultured from slants onto NA in order to obtain pure colonies. All plates were incubated aerobically at 37 °C (Gallenkamp, Cambridge, UK) for 18–24 h. Bacterial suspensions were prepared by transferring 2–3 colonies from the previously described plates into 0.85% saline. After mixing using a vortex mixer (Fisherbrand, Whrilmixer, Loughborough, UK), the turbidity of the suspensions was adjusted to achieve an optical density (OD) equivalent to McFarland 0.5 (approximately 10^5^–10^6^ cfu/mL) (DensiCHEK plus, BioMérieux, Marcy-l’Etoile, France).

A sterile cotton swab and a rotating turntable (Denley, Mast Diagnostics, Bootle, UK) were used to evenly inoculate three plates of DST (diagnostic sensitivity test) with the bacterial suspension. A sterile Pasteur pipette with a 6 mm diameter was used to make three wells in the agar in each plate. The sterile tip of the pipette was used to remove the plugs from the wells. Wells were filled with 100 μL of the neat oil. The volume of agar medium was approximately the same (24 mL) in all plates and was selected so that the oil in the well just touched the top surface of the medium where the microorganisms were inoculated. Each plate had three wells with three different oils. The type of oil in each well was chosen randomly but each plate had the essential oils of both grades (Hojari and Sha’bi). The plates were incubated at 37 °C for 18–24 h, and then the diameter of the growth inhibition zones were measured using a ruler in two sites to allow for non-uniform zones. Areas where bacterial colonies start to heap up around the well were not included in the measurement. The purity and the number of bacteria in each suspension were checked by diluting the bacterial suspension in peptone water and then spreading onto NA plates and incubating in the same conditions used for the tests.

The antifungal activity of oils against yeasts was performed in a similar way against bacteria but with some modifications. A preliminary experiment with yeasts showed that in PDA medium, large growth inhibition zones with yeasts that overlapped were produced if more than one well was made in the same plate; therefore, each plate contained only one well, and the experiment was performed in triplicate. Counts of yeast cells and the purity of inocula were confirmed as for bacteria. All plates were incubated at 37 °C aerobically for 48 h. For the molds, the antifungal activity of the essential oils was tested against spores. The spores were allowed to develop on PDA plates and then were harvested by washing with sterile distilled water. The suspension was mixed vigorously with a vortex mixer to obtain a homogeneous suspension of spores and adjusted to 10^5^–10^6^ spores/mL. Counts were made using a hemocytometer (Neubauer Improved, Darmstadt, Germany). The spore suspension was used as the inoculum, and the antifungal activity of the oils against molds was performed in the same way as described for the yeasts. A control plate for each mold was included, and all plates were incubated aerobically at 25 °C (Sanyo, Tokyo, Japan) for 48 h.

#### 2.4.3. Determination of the Minimum Inhibitory Concentrations (MICs) and Minimum Bactericidal/Fungicidal Concentrations (MBCs/MFCs)

A micro-well dilution method was used to determine the MIC of the essential oils against bacteria, yeasts and molds [5]. In brief, for each microorganism, oils that produced the largest inhibition zones (one from Hojari grade and one from Sha’bi grade) were selected to determine their MICs. 96-well microplates (Ser-wel, polystyrene, London, UK) with a u-shaped bottom were used. The microorganisms were prepared in the same way as for the screening test.

For bacteria, 90 μL of nutrient broth (NB) was added to nine wells. Then, doubling dilutions of the oil in NB were prepared in universal bottles and mixed vigorously using a vortex mixer (Fisherbrand, Whrilmixer, Loughborough, UK) in order to suspend the oil in the water-based NB. One hundred microliters of appropriate oil dilutions were transferred to their assigned wells in the microplate to make dilutions ranging from 50% to 0.20% (*v*/*v*). 10 μL of the bacterial suspension was added to all of the nine test wells that contained oils. For controls, 200 μL of NB was added to one well and served as a broth control; 190 μL of NB and 10 μL of bacterial suspension were added to one well to serve as a bacterial growth control. All tests were performed in triplicates. An oil control for all dilutions was included by adding 200 μL of the oil dilution (prepared in NB) to new wells. This was included to check if the oil contributed to the OD, and if so, its reading was subtracted from the test oil OD reading. All wells received a total volume of 200 μL. All microwell plates were sealed, shaken in an orbital shaker (Gallenkamp, Cambridge, UK) at 150 rpm for 1 min, and then incubated aerobically at 37 °C for 18–24 h.

After incubation, the absorbance was read at 620 nm using a microplate reader (ThermoLabsystems, Multiskan EX, Thermo Scientific, Vantaa, Finland). Wells with no turbidity and one previous well (higher oil dilution) were subcultured (10 μL) onto NA plates and incubated as stated before. Subculturing was performed to check the presence/absence of viable bacteria. Thus, the bactericidal or fungicidal concentration was defined as the lowest concentration of the oil that gave no growth after subculturing onto solid media under appropriate incubation conditions [28,29]. MIC was defined in two ways: (1) the lowest concentration of oil that showed no visible turbidity [28], which was considered as the visual MIC; and (2) the lowest concentration of essential oil required to decrease the absorbance or optical density (OD) of the microbial control by at least 90% [30]. The percent inhibition of each concentration of essential oil was calculated by comparing the average absorbance of the test wells with that of the growth control wells [20] using the following equation:(2)$$\%\;\mathrm{Inhibition}=\frac{{\mathrm{OD}}_{\mathrm{GC}}-{\mathrm{OD}}_{\mathrm{TO}}}{{\mathrm{OD}}_{\mathrm{GC}}}\times \;100$$
where GC = the growth control, and TO = the test oil.

The MICs and MFCs were determined for yeasts and molds using the same method as described for bacteria, using yeast malt broth (YMB) instead of NB, PDA instead of NA and incubating the yeasts at 37 °C for 48 h and the molds at 25 °C for 48 h.

### 2.5. Analysis of Frankincense Smoke Solid Particles

The determination of the size of particulate matter present in frankincense smoke was achieved using a microscopic technique. The method employed generating smoke by heating 5 g of ground frankincense placed on aluminum dish on a hot plate until complete burning and exposing a clean slide to this smoke. Slides were then examined using light microscope connected to a camera (Olympus BX51, Tokyo, Japan) and programmed so that the diameter of the solid particles can be determined. The view of slides exposed to frankincense smoke were compared to control slides that were not exposed to the smoke. Four samples of Hojari grade and four samples of Sha’bi grade were examined. For each samples, three slides were made, 20 fields were examined, and three particles in each field were measured for their largest and smallest diameters, and for their area. As the shape of the solid particles was irregular or non-spherical, the measured diameter was considered as the equivalent diameter [31].

### 2.6. Screening Antimicrobial Activity of Frankincense Smoke against Airborne Microbes

#### 2.6.1. Collection of Airborne Microorganisms

NA plates were opened in the lab, and the agar exposed and microorganisms allowed to deposit for two hours [32]. Subsequently the plates were incubated at 25 °C for 4 days. From this, a number of microorganisms with differing morphologies were purified and stored on NA slants for use as test organisms. These organisms were identified as three Gram-positive bacteria (B14, B17 and B18), one mold (M1) and one yeast (M7) based on their morphology and microscopic characteristics, with no further biochemical or genetic identification performed for the airborne isolates. Calibration curves of optical density (OD) at 470 nm and colony-forming units were prepared to determine the OD value for each specific isolate required to attain about 300 colonies growing on inoculated plates. Similarly, *S. aureus* (NCTC 6571) and *E. coli* (NCTC 10418) were used as QC bacteria to test the antimicrobial activity of frankincense smoke. They were subcultured onto NA plates and incubated at 37 °C for 24 h, and calibration curves were prepared, as previously mentioned.

#### 2.6.2. Exposure of Airborne Microorganisms to Frankincense Smoke

A method was developed to test the antimicrobial activity of frankincense smoke. To generate smoke, 25 g of ground Hojari white (HW) frankincense resin was burned to exhaustion (for about 20 min.) in an electric incense burner in a closed cupboard (dimensions: 96 × 54 × 35 cm). Three test plates inoculated with bacteria and three uninoculated NA plates were exposed to the smoke together inside the closed cupboard. The plates were placed into the cupboard covered whilst the resin was burning. Once the burning was complete and the smoke had been generated, the test plates were opened and left exposed to the smoke for two hours to make it possible for the active components in the smoke to contact the test microbes (the highest temperature and relative humidity were 27 °C and 60%, respectively). Three control plates were kept under the same conditions but closed. The control plates and the plates with exposed bacteria were incubated at 37 °C for 24 h. The three uninoculated plates that had been exposed to smoke were swabbed with the test bacteria and incubated at 37 °C for 24 h. Colony counts were determined for all plates. The experiment was repeated with three different resin samples.

The molds were grown on PDA plates, and a small circle of mycelium was cut using the base of a sterile Pasteur pipette and placed upside down onto a fresh PDA plate. Six test plates were exposed as described previously for bacteria, and three inoculated plates that were not exposed to the smoke were included as controls. The uninoculated PDA plates that had been exposed to smoke were inoculated with a piece of mycelium using the same method. The plates were incubated at 25 °C for 48 h. After incubation, the longest and shortest diameters of the mycelium were measured and compared to those of the controls. The percent inhibition of mold growth can be calculated from this equation:(3)$$\%inhibition=\frac{D_{C}-D_{T}}{D_{C}}\times 100$$
where *D_C_* = the diameter of mold colony on control plate, and *D_T_* = the diameter of mold colony on test plate.

### 2.7. Statistical Methods

Analysis of variance (ANOVA) was used to compare the growth inhibition zone diameters produced by each grade of frankincense oil, the MICs and MBCs/MFCs of the two grades, and to compare between the visual and the OD method used to determine the MICs. Excel 2016 was used to perform basic data analysis, and Data Disk 6.1 (Data Description, Inc., New York, NY, USA) was used to perform the statistical tests to identify any significant differences, which was considered as *p* < 0.05. Logarithmic transformation was applied for the dimensions of frankincense solid particles to enable data presentation.

## 3. Results

### 3.1. Yield and Chemical Composition of Essential Oils

The steam distillation process separated the oleo-gum resin of frankincense into its three main components: the essential oil; the gum fraction (recovered as a solid mixed with other impurities on the mesh); and the resin fraction, which was pure, yellow, non-water-soluble and fragile. Frankincense water (hydrosol) was also collected under the essential oil (Appendix A). The distilled fresh frankincense oil had a very fresh and perfume-like aroma. The percent yield (*w*/*w*) of oils and their chemical composition are presented in Appendix A. The mean yield from Sha’bi was slightly larger than from Hojari grade, but no significant difference was detected (*t*-test, *p* = 0.5269, α = 0.05). The HW3 and S1 samples gave similar highest percentages of oils (7.4%), while the HG sample gave the least (6.6%).

GC-MS revealed the presence of more than 90 chemical compounds in each frankincense sample. Only the most abundant 37 compounds were identified (Appendix A). Generally, the essential oil of Hojari and Sha’bi frankincense showed very similar chemical compositions, and the differences were mainly in the quantities of the chemical compounds. A small variation in the type of chemicals and their quantities was found in different samples belonging to the same grade. The major compound α-pinene was present in all samples at a level exceeding 50% of the total. Camphene, (+)-3-carene and 1-limonene were also present in all samples at considerable percentages (chemical structures are shown in Figure 1). Among the 37 compounds identified, 26 were present in all samples (compound reference **2**, **3**, **4**, **5**, **6**, **7**, **8**, **9**, **10**, **11**, **13**, **14**, **15**, **16**, **18**, **19**, **21**, **22**, **24**, **26**, **27**, **28**, **29**, **32**, **33** and **37**). Myrtenal was found in all samples except S4 and HW1. α-Copaene and δ-cadinene were found in all samples except S1 and HW2. *Allo*-Ocimene was found only in S4, S5, HW1, HW2 and HG. Germacrene A was found in S1, HW2 and HG samples. Five compounds were found only in one sample each, namely 4-Hydroxy-4-methyl-2-pentanone in S1, *trans*-(+)-carveol in HW2, germacrene-D in HW3, *allo-*aromadendrene in S4, and hedycaryol in HW1. Compounds **2** to **24** were monoterpenes, and compounds **25** to **37** were sesquiterpenes. No diterpenes were detected in any sample. Compounds **13** to **24** were oxygenated monoterpenes with the exception of *allo*-ocimene. Hedacaryol was the only oxygenated sesquiterpene identified.

### 3.2. Antibacterial and Antifungal Activity of Frankincense Oil

All bacteria, yeasts and the spores of molds were susceptible to Hojari and Sha’bi frankincense oils (Appendix A, Figure 1). For bacteria, the smallest inhibition zone was produced with *E. coli* using Sha’bi and Hojari oils (mean = 10.4 ± 1.3 and 10.4 ± 0.9 mm, respectively). The inhibition zones that were produced by the oils on *S. aureus* and *Bacillus* spp. were slightly larger than those produced on *E. coli* and *P. aeruginosa*. For fungi, the largest growth inhibition zone was produced with *S. cerevisiae* using both Sha’bi and Hojari oils (mean diameter = 37.8 ± 6.7 and 36.3 ± 8.2 mm, respectively) followed by *F. solani* with Sha’bi and Hojari oils (mean = 29.1 ± 4.1 mm and 28.2 ± 2.4 mm, respectively) (Figure 2). The smallest inhibition zone was produced with *A. flavous* using Hojari oil (mean = 10.4 ± 1.3 mm). *A. flavous* followed by *P. citrinum* were the most resistant molds to Sha’bi and Hojari oils. The yeast *S. cerevisiae* was slightly more susceptible than *C. albicans*. Regarding molds, the greatest inhibition zone was produced with *F. solani*.

S1 oil produced the largest inhibition zone with a mean diameter of 21.3 ± 10.3 mm (Appendix A). HW3 oil was the most active oil from the Hojari grade and produced inhibition zones with a mean diameter of 18.3 ± 10.4 mm. HG oil was the least active and produced the smallest inhibition zone with a mean diameter of 15.5 ± 7.4 mm. However, it produced a good inhibition zone with *F. solani* (Figure 3)

Statistically, the grade of frankincense had no significant effect on the diameter of the inhibition zones (ANOVA, *p* = 0.0708, α = 0.05). However, the diameter of inhibition zones was significantly influenced by the type of microorganisms (ANOVA, *p* = 0.0001, α = 0.05). There was no significant interaction between the grade of frankincense (Hojari and Sha’bi) and the type of microorganism on the diameter of inhibition zones (ANOVA, *p* = 0.3980, α = 0.05).

### 3.3. MICs and MBCs/MFCs of Frankincense Oil

Frankincense oil at its highest concentrations (50, 25, and 12.5%; 500,000, 250,000, and 125,000 µg/mL) dissolved the wells of the microtiter plates, confounding the optical density readings [33]. The absorbance results at these concentrations were not further analyzed. The absorbance of oil controls made in NB was not found to be different from the NB control. Therefore, the absorbance of test wells containing NB did not need correction. However, the absorbance of oil controls that were made in YMB differed from the YMB controls, but only at oil concentrations of 6.25 and 3.13% (62,500 and 31,300 µg/mL). The absorbance of these wells was adjusted by subtracting the corresponding absorbance of oil controls. The results of the absorbance measurements are shown in Appendix A. The absorbance values were used to calculate %inhibitions in the growth of different microorganisms induced by Hojari and Sha’bi frankincense oil (Table 1). The percent inhibitions were very low at lower oil concentrations (0.78, 0.39 and 0.20%). However, in general, all oils reduced the growth of test microorganisms by more than 80% when at a concentration of 6.25% (62,500 µg/mL).

Generally, the growth inhibition of the microorganisms by frankincense oil followed the pattern shown in Figure 4 for *F. solani*. The pattern could be divided into three areas: an area with little growth inhibition at lower oil concentrations, followed by a region in which increasing oil concentrations produced an exponential increase in growth inhibition until a plateau was finally reached.

The results of the MICs and MBCs/MFCs of frankincense oil are shown in Table 2. An MIC value of 12.5% (125,000 µg/mL) was determined visually with *A. flavous*, *A. ochraceus* and *P. citrinum* (Table 2). The lowest MIC (1.56%; 15,600 µg/mL) was found with *E. coli*, *S. cerevisiae*, *F. solani* and *A. niger* (Table 2). Only the type of microorganism was found to influence the MIC of oils significantly (ANOVA, *p* = 0.0335, α = 0.05). The MIC was not significantly affected by the grade of frankincense or the method of determination (ANOVA, *p* = 0.3156 and 0.1085, respectively, α = 0.05). No significant interaction on the MIC values was detected between the method and the grade (ANOVA, *p* = 0.3739, α = 0.05), the method and the type of microorganism (ANOVA, *p* = 0.8523, α = 0.05) or the grade and the type of microorganism (ANOVA, *p* = 0.5995, α = 0.05).

In the literature, there is a lack of information regarding the bactericidal and fungicidal concentrations of frankincense oil. In the present study, MBC/MFC values, when found at the highest oil concentrations (50, 25, and 12.5%; 500,000, 250,000, and 125,000 µg/mL), were reported in this study, although they should be confirmed using a resistant material for the wells. It was possible to determine the MIC when it was higher than 6.25% (62,500 µg/mL) only visually. The highest MFCs (≥50%; ≥500,000 µg/mL) were found with all of the molds except *F. solani,* which was more susceptible (MFC = 3.13%; 31,300 µg/mL) to both oils (Table 2). *E. coli* was the most susceptible bacterium and showed an MBC of 3.13%, or 1.56% compared to the MBC of 12.5% with all other bacteria. The MBC/MFC was not significantly affected by the grade of frankincense (ANOVA, *p* = 0.0865, α = 0.05), while the type of microorganism significantly affected the MBC/MFC (ANOVA, *p* = 0.0001, α = 0.05).

### 3.4. Dimensions of Frankincense Smoke Solid Particles

Average results of analysis of frankincense smoke solid particles are presented in Figure 5 for the first time. The equivalent diameters varied greatly, with the shortest diameter being 0.75 µm and the largest one being 2287 µm. The appearance of some solid particles is shown in Figure 6.

### 3.5. Antimicrobial Activity of Frankincense Smoke

The smoke of frankincense Hojari green completely inhibited the growth of the control bacteria (*S. aureus* and *E. coli*), the airborne bacteria (B17) and the airborne mold (M1) (Table 3, Figure 7). The Hojari white sample also produced 100% inhibition on the growth of all airborne bacteria (B14, B17 and B18), and the yeast (M7). It also reduced the growth of *S. aureus* (NCTC 6571) and *E. coli* (NCTC 10418) by 66 and 73%, respectively (Table 3).

## 4. Discussion

The Hojari and the Sha’bi frankincense produced good yields of essential oils of about 7% for both grades after distillation for 5 h. These yields were greater than the hydro-distilled oil of the same species (5.5%) [25] and the steam-distilled oil of *B. carterii* (3%) [34]; however, they were less than what was obtained by another study for *B. sacra* hydro-distilled oil (8.9–12.0%) [13]. The differences in yields could be attributed to many factors, such as the plant species, the freshness of the oleo-gum resin, the distillation method, and the harvest and storage conditions.

In agreement with a previous study [13], the monoterpene α-pinene was the major compound in *B. sacra* essential oil, and its level exceeded 50% in all samples. In contrast, *E*-β-ocimene (32.3%) and limonene (33.5%) were reported to be the major compounds in the hydro-distilled oil of *B. sacra* in another study [25]. The other compounds that were found in considerable percentages in this study were the monoterpenes camphene, (+)-3-carene, 1-limonene and 1-β-pinene, and the sesquiterpene β-elemene, which were detected at lower percentages by other investigators [25]. Differences in the chemical composition is unlikely to be due to factors related to the climate conditions or geographical locations of the trees as this study demonstrated the similarities in the chemical composition of the Hojari and the Sha’bi oils. However, the method of oil extraction might be a contributing factor as well as the harvest time during the year and the incision number the frankincense was collected from.

Similarly, a previous study [35] reported only quantitative variations in the triterpenoid fraction (part of the resin) of the *B. sacra* oleo-gum resin that was collected from different locations in Dhofar (Oman), including the Al-Mughsayl, Wadi Adonib, Hasik, Wadi Dawkah and Salalah market. α-boswellic acid, β-boswellic acid, 3-O-acetyl-α-boswellic acid, 3-O-acetyl-β-boswellic acid, 11-keto- α-boswellic acid, 11-keto- β-boswellic acid and 11-keto-3-O-acetyl-α-boswellic acid were recognized in all samples, and it was concluded that no significant difference was detected in the composition of the triterpenoid fraction. The quantitative dissimilarities were attributed to the collection time rather than to the location of the trees. The results of this study also indicate that the essential oil of the Sha’bi and the Hojari Omani frankincense are not different in their chemical composition but have quantitative variation.

Mono- and sesquiterpenes cannot be considered as characteristic of frankincense because of their frequent occurrence in other plants. Nevertheless, diterpenes and triterpenes can be used as biomarkers or fingerprints to identify some species of *Boswellia* [15]. Similar to a previous study [25], no diterpenes were detected in this study. However, diterpenes have been reported in the hydro-distilled oil of *B. carterii* [36,37]. Oxygenated compounds in the essential oil contribute more to the total fragrance and give the oil more stability than nonoxygenated compounds and thus improve the quality of the essential oils [36]. Several oxygenated monoterpenes (compounds 13 to 24 except *allo*-ocimene and one oxygenated sesquiterpene (hedacaryol)) were identified, similar to another study [25].

Unlike the disc diffusion method, the well technique allows for the use of a larger volume of the test compound, thus allowing for the better detection of its inhibitory activity, especially if the active chemical components are present at low concentrations. Also, the disc diffusion method is more prone to gives false negative results than the well technique due to volatility of the essential oils [38,39]. It was reported that 60% of essential oil derivatives examined before 1977 were active against fungi, while only 30% were active against bacteria [40]. All organisms tested in this research were susceptible to frankincense essential oil (Figure 2) as all inhibition zones were ≥7 mm [41]. The inhibitory activity of frankincense oil in this study indicates that the active components in the steam-distilled essential oil of *B. sacra* have broad-spectrum antimicrobial activity. Utilizing the well technique in this study (Figure 2), *S. aureus* and *Bacillus* spp. were slightly more susceptible to Sha’bi and Hojari frankincense oil than *E. coli* and *P. aeruginosa*. This agrees with a previous study [42], in which the antibacterial activity of 15 medicinal plants using a disc diffusion method was detected. Their results showed *B. subtilis* to be more sensitive than *S. aureus* and *P. aeruginosa*. Likewise, using the well technique in this study, *Bacillus* spp. was the most susceptible bacterium to the Sha’bi and the Hojari oils.

Using an agar dilution method, an ethanol extract of a commercial powder of *B. serrata* showed no activity against *C. albicans*, though it inhibited the growth of *B. subtilis* and *S. aureus* [43]. In this study, the essential oils of Sha’bi and Hojari frankincense were active against *C. albicans*. *S. cerevisiae* was one of the most susceptible microorganisms using both the well diffusion method and the micro-well dilution method. Gram-negative bacteria have more physicochemical complexity in their cell wall than Gram-positive bacteria and yeasts [8]. The outer membrane of Gram-negative bacteria contains lipopolysaccharide (LPS) [44], which increases the bacterial surface hydrophilicity [45]. This blocks the penetration and accumulation of essential oil into and on the cell membrane [44]. However, Gram-negative bacteria have porin proteins in the outer membrane, which create channels that allow for the restricted passage of small molecules. Sterols that are present in fungi have not been found to confer resistance against antimicrobial agents [8].

The spores of *Aspergillus*, *Penicillium* [46] (p. 10), and *Alternaria* are unwettable [47], while those of *Fusarium* are wettable [46] (p. 10). Because tests were performed in water-based media, the wettable spores might have allowed for better penetration and action of the active components. This might be the reason why *F. solani* in this study was more susceptible to oils than other molds. Other investigators [48] tested the antifungal activity of frankincense oil (*B. carterii*) against seven species pathogenic to rice, including *A. flavus*, *Alternaria brassicicola*, *Fusarium moniliforme*, *Fusarium proliferatum*, *Bipolaris oryzae*, *Rhizoctonia solani* and *Pyricularia arisea*. The oil at a concentration of 2% (*v*/*v*) was able to inhibit spore germination and mycelia growth of all pathogenic fungi, with the strongest action against *F. moniliforme*.

Freshness of the oil might also be a factor that contributes to its antimicrobial activity. In this study, S1 and HW3 were the freshest and showed slightly more activity than other oils. However, distillation was performed over a short period of time, and all the oil samples were tested simultaneously. The visual method allowed for the determination of MICs for all oils, but the OD method could not be used when the MIC was higher than 6.25% (*v*/*v*) because the wells dissolved. Other researchers [20] used polystyrene microtiter plates to test anti-biofilm activity of *B. papyrifera* steam-distilled essential oil at concentrations ranging from 25% (*v*/*v*) to 0.75% (*v*/*v*) and did not report if the oil dissolved the wells. In this study, at high concentrations of oils (12.5, 25 and 50%, *v*/*v*), the wells appeared cloudy and were sticky and soft when touched indicating that the integrity of the plastic had been destroyed, i.e., the oils had dissolved the plastic [33]. However, this effect was reduced at lower oil concentrations. Some compounds, such as cymene, terpinene, limonene, and phellandrene, which are frequently found in essential oils, including frankincense oil, were previously reported to be good solvents for polystyrene foams [33]. It will be interesting to identify which components in frankincense oil are responsible for dissolving different types of plastics and determine their applicability.

In this study, the MICs of all bacteria except *E. coli* were similar. Species variation in each bacterial group of Gram-positive and Gram-negative bacteria may lead to variation in their susceptibility to antimicrobial agents. For instance, in this study, *E. coli* had lower MIC than *P. aeruginosa*. Likewise, our previous study using the same strains of *E. coli* and *P. aeruginosa* found lower MIC results for *E. coli* (0.78%) than *P. aeruginosa* (1.56%) with both clove and thyme essential oils [27]. An ethanol extract of powder from *B. serrata* was shown to exhibit a higher MIC (100 μg/mL) with *S. aureus* than with *B. subtilis* (2 μg/mL) [43]. All molds except *F. solani* had similar MICs and MFCs. Fungal spores were more resistant to frankincense oil than the vegetative cells of bacteria and yeasts (MFC of 50%, *v*/*v*). Other researchers [45] considered the estimation of the bactericidal concentrations of essential oils to be more precise than the agar well technique. They found inhibition zones produced by rosemary essential oil against *S. aureus* and *Campylobacter jejuni* to be 5.9 and 9.3 mm, respectively. However, their bactericidal concentrations were 0.1% and >1% for *S. aureus* and *C. jejuni*, respectively. While testing the antimicrobial activity of the essential oil of *B. rivae* against *C. albicans*, an inhibition percentage of 81.5% was produced with 0.3% (*v*/*v*) [20]. In this study, higher oil concentrations of 3.1% (*v*/*v*) and 6.3% (*v*/*v*) with Sha’bi and Hojari oils, respectively, were required to reduce the growth of *C. albicans* to 95.4% and 96.2%, respectively. Thus, frankincense oil used in this study can be considered to have moderate antimicrobial activity. High MIC values might indicate that the active components are present in the extract at low concentrations as a result of the extraction method [5]. This supposes that detecting the antimicrobial activity of the crude extracts is essential, regardless of their potency.

In general, the pattern of the growth inhibition of the most tested microorganisms by frankincense oil was similar to the pattern shown in Figure 4. There was little growth inhibition at lower oil concentrations, followed by a dramatic increase in the growth inhibition as the oil concentrations were increased, until a plateau was reached when the growth inhibition was nearly constant. A similar trend was described previously [49], demonstrating that plotting an inhibitor concentration and its inhibition potential gives a curve with a sigmoid shape.

Regarding molds, clear cut results can be obtained for MICs using the method described in this study by checking spore germination microscopically. The presence/absence of germinated spores can be seen clearly in the microplate wells using light microscopy. However, some researchers [48] preferred to count the number of germinated spores after inoculating a spore suspension onto media smeared on microscope slides and subsequently exposed to different concentrations of frankincense oil. Then, the percent reduction in spore germination could be calculated.

No significant difference was found between the two grades of frankincense essential oils (Hojari or Sha’bi) in terms of the diameter of growth inhibition zones, MIC or MBC/MFC results. This is expected as the chemical composition of the oils of the two grades was similar. However, the diameter of the inhibition zones, MICs and MBCs/MFCs were significantly different for different microorganisms. This is also expected as different microorganisms differ in their genetic makeup and respond differently to an antimicrobial agent.

In this study, α-pinene was a major chemical compound found in all oil samples; it might be the principle active chemical and thus should be investigated further to determine its antimicrobial potential as a pure compound and its mechanism of action. Essential oils are known to interact with cell membranes, and this can affect the function of different membrane components, such as receptors, enzymes, transport systems, and ion channels. It was reported that synergy between monoterpenes, like pinene, and sesquiterpenes, such as β-caryophyllene, can occur. The compounds α-humulene and myrcene (which were present in the oil used in this study) had good antimicrobial activity against some Gram-positive and Gram-negative bacteria as well as some yeasts [50]. Thus, the essential oil used in this study contained chemical compounds that are known to have activities against bacteria, yeasts and molds, which might explain its broad-spectrum antimicrobial activity. However, the purification and identification of specific bioactive chemical compounds in Hojari and Sha’bi frankincense oils require further investigation.

This study is thought to be the first study that provides a detailed description of the solid particles of frankincense smoke and their antimicrobial activity. The results showed that frankincense smoke contains fine particles, in which the shortest diameter ranged from 0.8 to 1.6 µm for Sha’bi samples and from 1.3 to 2.0 µm for Hojari samples. Thus, it is possible for these fine particles to find their way throughout the respiratory system and possibly affect susceptible individuals. Likewise, other investigators [51] characterized the emission of particulate matter by burning 23 different types of incense, including a cone made in Mexico from frankincense and myrrh. They concluded that burning incense emits fine particulate matter in large quantities that may pose a health risk to people inhaling them. Another study was conducted to investigate whether home exposure to Arabian incense (bakhour) contributes to the prevalence of asthma and/or triggers its symptoms in 10-year-old Omani schoolchildren. Arabian incense burning was found to be a common trigger of wheezing among asthmatic children. However, no significant association between incense burning and the prevalence of asthma was found [21]. Likewise, incense burning was found to be a precipitating factor for asthma attacks in Qatari children but not a risk factor for chronic obstructive lung disease in a case–control study of adults in Saudi Arabia [51].

To elucidate the chemical composition of the pyrolysate or the smoke of frankincense, some researchers burnt frankincense (*B. carterii*, *B. rivae*, *B. frereana*, *B. serrata* and *B. neglecta*) on red-hot charcoal, and then the pyrolysate was analyzed by GC-MS. The smoke of *B. carterii* was found to contain volatiles, such as cembrene A; cembrene C; incensyl acetate and verticilla 4(20),7,11-triene incensole. *p*-Camphorene, *m*-camphorene, cembrenol, cembrene A, sabinene, α-cubebene, limonene, γ-muurolene and β-bourbonene were found in the smoke of *B. serrata*. The dimer of α-phellandrene and lupeol were revealed in the smoke of *B. frereana*, and amyrin derivates were found in the smoke of *B. neglectae* and *B. rivae* [12].

Some researchers [52] isolated two compounds, namely 2,9-dimethylpicene and 1,2,4a,9-tetramethyl-1,2,3,4,4a,5,6,14b-octahydropicene, from the *n*-hexane extract of the smoke-saturated water of Omani frankincense and found that they had anticancer activity against MDA-MB-231 breast cancer cells. This study demonstrated the antimicrobial activity of frankincense smoke against *S. aureus*, *E. coli*, and airborne bacteria, yeast, and mold. The Hojari green sample showed 100% inhibition of the tested microbes with which these preliminary findings provided an insight into exploring the potential of developing novel products from frankincense smoke with various applications. For instance, the smoke itself may be used for smoking foods that are usually intended for producing smoked foods as it is being used for sanitizing water containers in some countries [16]. This may contribute to killing microbes and giving distinct flavor to the smoked foods. The Flavor Extract Manufacturer’s Association (FEMA) gave frankincense general recognition as safe (GRAS) status (NO. 2765) as a flavor agent. In the food industry, frankincense is included in the production of beverages, candies, chewing gums, gelatins, nut products, puddings, and canned vegetables. In Saudi Arabia, about 500 tons of Somali frankincense are imported for chewing gum manufacturing, while a similar amount is used for burning in the home [16]. In Oman, recently, many products have appeared in the market with frankincense having been added, including the special Omani sweet (halwa) and various dairy products, such as milk, yoghurt and ice cream. Frankincense is added to food for its health benefits. However, it is important to determine its effect on gut microbiota and probiotics intended to deliver live cultures to the gut, taking into consideration different factors such as the concentration of frankincense and the type of its formulation. Moreover, further investigations can target and isolate the active compounds from frankincense smoke, which can be used as antimicrobial agents with various applications in the food industry, pharmaceuticals, and cosmetics. Investigating the potential of using frankincense smoke to fumigate animal cages as opposed to chemical disinfectants or alternately will remain a hot topic. This might contribute to the control of the presence and transfer of animal pathogens and their reach to foods and humans. Frankincense smoke is also known to repel insects. The efficiency of using filters to generate clean, safe, yet efficient sanitizing smoke can be tested in the future.

## 5. Conclusions

Frankincense oil possessed broad-spectrum antimicrobial activity against bacteria, yeasts and molds that varied significantly in intensity according to the type of microorganism. *S. cerevisiae* and *F. solani* were the most susceptible organisms. The similarity in the antimicrobial activity of the Sha’bi and the Hojari oils is attributed to their similar chemical composition, which indicates that either grade can be used to extract the essential oil for subsequent use in the food industry and medicine, although further investigation is required in relation to specific applications. Although frankincense contained fine solid particles that may pose a health risk for some individuals, the smoke proved its antimicrobial activity against *S. aureus*, *E. coli* and airborne bacteria, yeast, and mold, and thus, products may be developed from it for sanitation purposes or other uses.

## Figures and Tables

**Figure 1 foods-12-03442-f001:**
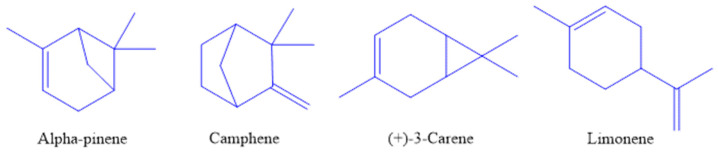
Chemical structures of the major compounds found in the essential oil of Sha’bi and Hojari frankincense. Drawn using ChemDraw Ultra (1985–2004, CambridgeSoft Corporation, Cambridge, MA, USA).

**Figure 2 foods-12-03442-f002:**
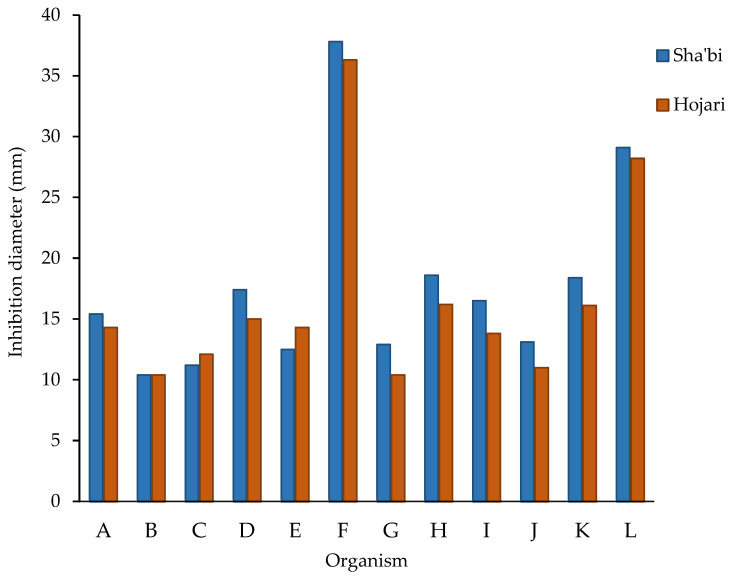
Means of the growth inhibition zones produced by Hojari and Sha’bi oils against 12 microorganisms. A: *S. aureus*, B: *E. coli*, C: *P. aeruginosa*, D: *Bacillus* spp., E: *C. albicans*, F: *S. cerevisiae*, G: *A. flavous*, H: *A. ochraceus*, I: *A. niger*, J: *P. citrinum*, K: *A. alternata* and L: *F. solani*.

**Figure 3 foods-12-03442-f003:**
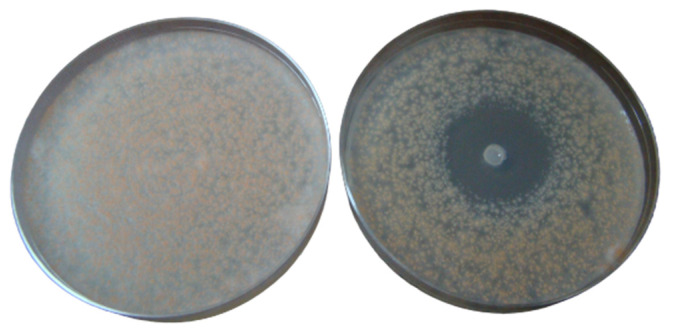
Inhibition of *Fusarium solani* with Hojari green frankincense oil. **Left plate**: control; **right plate**: test.

**Figure 4 foods-12-03442-f004:**
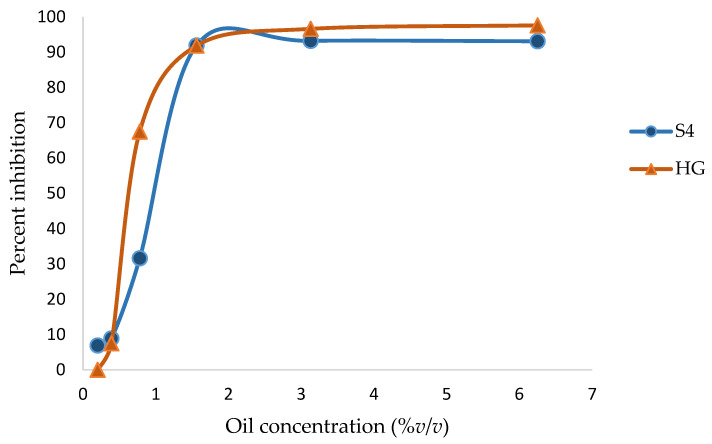
Percent inhibition of the growth of *F. solani* by Sha’bi 4 (S4) and Hojari green (HG) frankincense oils.

**Figure 5 foods-12-03442-f005:**
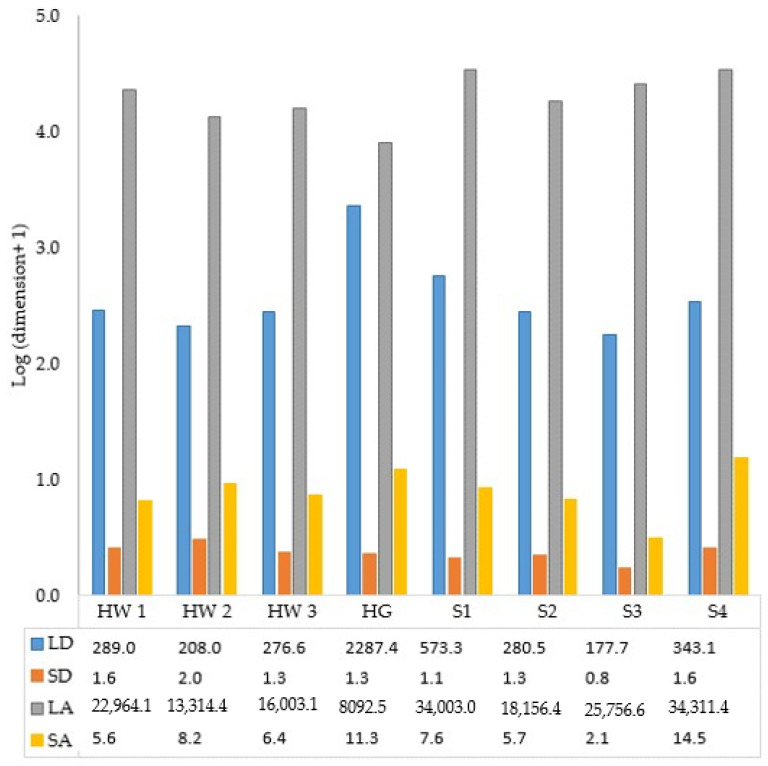
Dimensions (LD: longest diameter, SD: shortest diameter, LA: largest area, and SA: smallest area in µm) of frankincense smoke solid particles. Data table shows the actual results of the log_10_ transformed data. S1, S2, S3, S4, S5: Sha’bi samples 1,2,3,4 and 5, respectively. HW1, HW2, HW3: Hojari white samples 1, 2, and 3, respectively. HG: Hojari green.

**Figure 6 foods-12-03442-f006:**
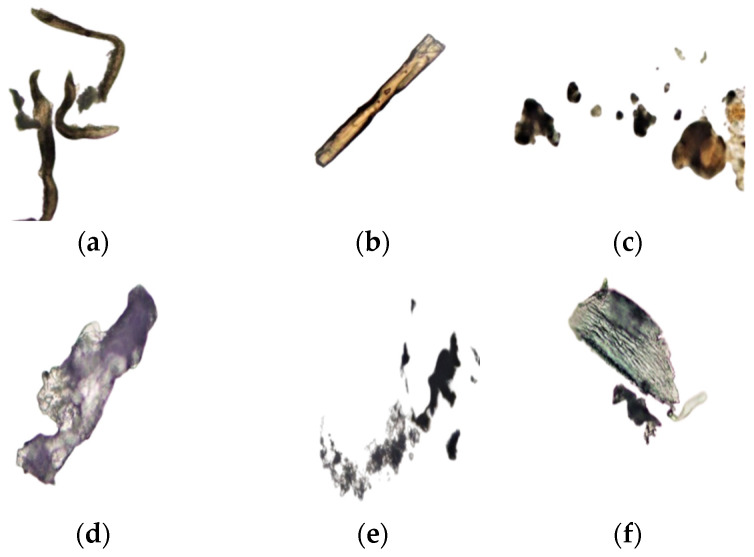
Microscopic (×40) appearance of some frankincense smoke solid particles according to the grade: (**a**): Hojari green, (**b**,**c**): Hojari white, (**d**–**f**): Sha’bi.

**Figure 7 foods-12-03442-f007:**
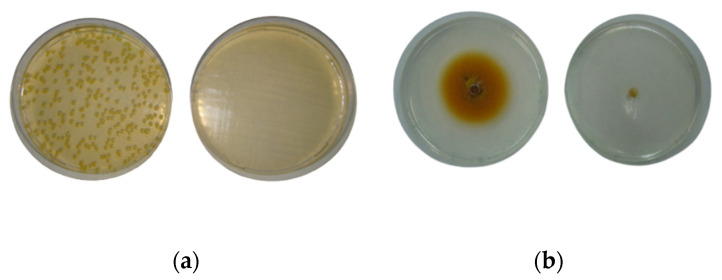
Inhibitory effect of frankincense smoke on the growth of airborne isolates: (**a**) bacteria, (**b**) mold. **Left plate**: control; **right plate**: test.

**Table 1 foods-12-03442-t001:** Percent reduction in OD_620_ of 12 microorganisms caused by Sha’bi and Hojari oils (excluding concentrations that dissolved wells).

Organism	Oil	Oil Concentration (% *v*/*v*; µg/mL between Brackets)					
		6.25 (62,500)	3.13 (31,300)	1.56 (15,600)	0.78 (7800)	0.39 (3900)	0.20 (2000)
*S. aureus*	S1	88.0 ± 1.9	65.8 ± 19.0	28.0 ± 28.7	13.4 ± 32.6	10.2 ± 31.0	11.7 ± 30.8
	HW3	87.7 ± 0.9	59.1 ± 10.6	34.7 ± 8.8	24.8 ± 10.2	23.7 ± 11.1	22.8 ± 13.5
*E. coli*	S1	90.8 ± 1.1	90.5 ± 3.5	92.6 ± 0.5	49.4 ± 1.7	18.4 ± 3.0	6.0 ± 5.7
	HW1	91.8 ± 0.8	92.7 ± 0.4	91.5 ± 2.1	32.0 ± 6.4	16.3 ± 6.7	13.5 ± 4.6
*P. aeruginosa*	S3	83.4 ± 2.7	5.3 ± 6.0	2.4 ± 4.4	−0.8 ± 5.6	−3.1 ± 4.0	−1.3 ± 4.3
	HW1	89.9 ± 1.1	9.1 ± 7.5	7.5 ± 5.9	5.2 ± 6.3	2.5 ± 6.3	3.4 ± 6.1
*Bacillus* spp.	S1	83.6 ± 0.5	5.9 ± 3.7	7.6 ± 11.5	11.2 ± 7.2	9.6 ± 4.7	7.8 ± 5.7
	HW3	81.2 ± 6.0	34.4 ± 11.2	34.0 ± 11.5	35.7 ± 9.9	33.6 ± 7.1	38.4 ± 5.8
*C. albicans*	S1	95.3 ± 0.3	95.4 ± 0.2	17.9 ± 8.8	6.3 ± 4.9	1.2 ± 6.7	3.7 ± 4.3
	HW2	96.2 ± 0.6	38.8 ± 23.8	30.3 ± 6.2	17.4 ± 8.8	9.9 ± 5.5	9.6 ± 3.0
*S. cerevisiae*	S1	91.6 ± 1.7	94.3 ± 0.3	92.5 ± 1.9	31.7 ± 14.6	11.7 ± 3.6	9.4 ± 5.0
	HW3	93.9 ± 0.7	94.8 ± 0.3	18.0 ± 22.0	51.0 ± 3.6	18.7 ± 12.1	22.2 ± 47.2
*A. flavous*	S4	89.3 ± 1.1	83.6 ± 2.0	65.5 ± 6.3	46.6 ± 11.3	37.3 ± 11.1	11.7 ± 7.8
	HW2	83.7 ± 3.5	81.1 ± 6.4	72.3 ± 9.6	66.1 ± 7.7	37.2 ± 23.0	38.0 ± 25.0
*A. ochraceus*	S3	83.0 ± 1.3	65.9 ± 15.5	69.3 ± 1.2	61.0 ± 1.5	39.4 ± 8.1	−8.5 ± 29.9
	HW3	60.1 ± 16.0	61.8 ± 9.6	49.0 ± 15.7	23.8 ± 19.7	−0.7 ± 16.4	20.7 ± 14.7
*A. niger*	S1	92.5 ± 2.6	88.4 ± 6.0	90.0 ± 3.1	83.2 ± 8.5	26.7 ± 22.5	−34.6 ± 34.8
	HW2	90.6 ± 2.7	90.0 ± 8.7	79.7 ± 36.9	27.4 ± 30.6	−9.0 ± 9.7	−35.9 ± 18.6
*P. citrinum*	S4	93.5 ± 2.7	77.2 ± 8.7	47.4 ± 36.8	59.1 ± 30.6	12.2 ± 9.7	−19.7 ± 18.6
	HW3	75.0 ± 2.1	63.1 ±6.0	50.0 ± 17.7	14.2 ± 5.3	17.7 ±8.0	−11.3 ± 9.7
*A. alternata*	S4	84.9 ± 7.3	72.6 ± 10.2	56.7 ± 14.5	56.1 ± 22.1	32.5 ± 32.3	2.2 ± 6.3
	HW2	82.3 ± 9.8	78.7 ± 6.5	44.3 ± 12.3	50.7 ± 8.3	0.9 ± 41.3	−59.4 ± 37.5
*F. solani*	S4	93.1 ± 0.7	93.2 ± 0.2	91.9 ± 1.2	31.6 ± 3.9	8.9 ± 2.9	6.9 ± 0.7
	HG	97.6 ± 0.9	96.6 ± 0.3	91.8 ± 1.3	67.5 ± 12.8	7.5 ± 18.3	0.0 ± 1.1

S1, S2, S3, S4, S5 = Sha’bi samples 1, 2, 3, 4 and 5, respectively. HW1, HW2, HW3 = Hojari white samples 1, 2, and 3, respectively. HG = Hojari green.

**Table 2 foods-12-03442-t002:** MIC (visual and OD method) and MBC/MFC (% *v*/*v*; µg/mL between brackets) of Sha’bi and Hojari frankincense oil against 12 microorganisms.

Microorganism	Oil Sample	MIC		MBC/MFC
		Visual	OD_620_	
*S. aureus*	S1	6.25 (62,500)	>6.25 (>62,500)	12.50 (125,000)
	HW3	6.25 (62,500)	>6.25 (>62,500)	12.50 (125,000)
*E. coli*	S1	1.56 (15,600)	1.56 (15,600)	1.56 (15,600)
	HW1	1.56 (15,600)	1.56 (15,600)	3.13 (31,300)
*P. aeruginosa*	S3	6.25 (62,500)	>6.25 (>62,500)	12.50 (125,000)
	HW1	6.25 (62,500)	6.25 (62,500)	12.50 (125,000)
*Bacillus* spp.	S1	6.25 (62,500)	>6.25 (>62,500)	12.50 (125,000)
	HW3	6.25 (62,500)	>6.25 (>62,500)	12.50 (125,000)
*C. albicans*	S1	3.13 (31,300)	3.13 (31,300)	3.13 (31,300)
	HW2	6.25 (62,500)	6.25 (62,500)	6.25 (62,500)
*S. cerevisiae*	S1	1.56 (15,600)	1.56 (15,600)	1.56 (15,600)
	HW3	3.13 (31,300)	3.13 (31,300)	3.13 (31,300)
*A. flavous*	S4	12.50 (125,000)	>6.25 (>62,500)	50.00 (500,000)
	HW2	12.50 (125,000)	>6.25 (>62,500)	50.00 (500,000)
*A. ochraceus*	S3	12.50 (125,000)	>6.25 (>62,500)	50.00 (500,000)
	HW3	12.50 (125,000)	>6.25 (>62,500)	>50.00 (>500,000)
*A. niger*	S1	6.25 (62,500)	1.56 (15,600)	50.00 (500,000)
	HW2	6.25 (62,500)	3.13 (31,300)	50.00 (500,000)
*P. citrinum*	S4	12.50 (125,000)	6.25 (62,500)	50.00 (500,000)
	HW3	12.50 (125,000)	>6.25 (>62,500)	50.00 (500,000)
*A. alternata*	S4	6.25 (62,500)	>6.25 (>62,500)	>50.00 (>500,000)
	HW2	6.25 (62,500)	>6.25 (>62,500)	>50.00 (>500,000)
*F. solani*	S4	1.56 (15,600)	1.56 (15,600)	3.13 (31,300)
	HG	1.56 (15,600)	1.56 (15,600)	3.13 (31,300)

S1, S3, and S4 = Sha’bi samples 1, 3, and 4, respectively. HW1, HW2, HW3 = Hojari white samples 1, 2, and 3, respectively. HG = Hojari green.

**Table 3 foods-12-03442-t003:** Effect of frankincense smoke on the growth of airborne and control microorganisms.

Microorganism	Mean Plate Colony Counts ± SD			
Control Bacteria	Hojari White		Hojari Green	
	Controls	Tests	Controls	Tests
*S. aureus* (NCTC 6571)	29.3 ± 16.6	10 ± 1	380.7 ± 155.5	ND
*E. coli* (NCTC 10418)	40.7 ± 14.5	11.0 ± 4.6	320.7 ± 152.0	ND
Air isolates				
B14 (Gram + bacteria)	203.3 ± 106.4	ND	-	-
B17 (Gram + bacteria)	160.7 ± 12.9	ND	431.67 ± 133.6	ND
B18 (Gram + bacteria)	225.0 ± 61.3	ND	-	-
M7 (yeast)	280.3 ± 89.5	ND	-	-
	Growth diameter (mm)			
M1 (mold)	-	-	27	0

ND: not detected; -: not tested. SD = standard deviation.

## Data Availability

The data presented in this study are available in this article.

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
