# Peer review of "Antimicrobial Activity of Frankincense (Boswellia sacra) Oil and Smoke against Pathogenic and Airborne Microbes"

_foods, 2023, doi:10.3390/foods12183442_

Round 1
Reviewer 1 Report
Dear authors.
I consider that the manuscript needs to be improved:
Abstract
Include the method to determine the smoke's particle analysis and antimicrobial evaluation.
Introduction
The content of this section can be improved. Several studies in the literature determine the chemical profile and antimicrobial activity of Boswellia sacra. However, the relationship or differences between these studies is not established. The study should be justified more clearly.
Abers, M., Schroeder, S., Goelz, L., Sulser, A., St Rose, T., Puchalski, K., & Langland, J. (2021). Antimicrobial activity of volatile substances in essential oils. BMC Medicine & Complementary Therapies, 21(1), 1-14.
Di Stefano, V., Schillaci, D., Cusimano, M. G., Rishan, M., & Rashan, L. (2020). In vitro antimicrobial activity of Boswellia sacra frankincense oils grown in different locations in the Dhofar region (Oman). Antibiotics, 9(4), 195.
Al-Harrasi, A. and Al-Saidi, S. (2008). Phytochemical analysis of Boswellia sacra (Omani Luban) botanically certified oleogum resin essential oil. Molecules, 13(9), 2181-2189.
Material and methods
2.1. Include identification voucher.
2.6. The method is validated. Are there any other studies that support the effectiveness of this method?
Results
How does this study differ from the other reported studies? What is new information reported?
Chemical characterization. The results of the characterization are consistent with the literature. In this sense, consider that originality can be improved. Highlight if novel results were found.
Antimicrobial activity
Report the results as µg/mL, to perform comparative analyses. Same previous comment, highlight if novel results were found.
3.2. First, describe the results for bacteria and later yeast and fungi.
3.3. Describes more precisely the most relevant results. The most active treatments.
3.5. To which bacterium, fungus, or yeast does the sample b14, b17, b18, m7 and m1 correspond ? Because they did not carry out biochemical or molecular tests.
Discussion
Line 561-562-The results in the table show that E. coli was more susceptible. Is there a typing error? It does not agree with what is discussed in the table.
Line 571-577. P. aeruginosa and E. coli are Gram -, explain why one is more susceptible than the other. Are the antimicrobial found results important? Can they be classified as extracts with high antimicrobial potential? Compare with natural product classifications.
Describe the possible mechanism of action of the treatments evaluated based on the main compounds.
Author Response
Point 1: Abstract: Include the method to determine the smoke's particle analysis and antimicrobial evaluation.
Response 1: This was included in the abstract as follows:
“A microscopic technique was used to determine the size of frankincense smoke solid particles. Microbes were exposed to frankincense smoke to test their susceptibility to the smoke.”
Point 2: Introduction
The content of this section can be improved. Several studies in the literature determine the chemical profile and antimicrobial activity of Boswellia sacra. However, the relationship or differences between these studies is not established. The study should be justified more clearly.
Abers, M., Schroeder, S., Goelz, L., Sulser, A., St Rose, T., Puchalski, K., & Langland, J. (2021). Antimicrobial activity of volatile substances in essential oils. BMC Medicine & Complementary Therapies, 21(1), 1-14.
Di Stefano, V., Schillaci, D., Cusimano, M. G., Rishan, M., & Rashan, L. (2020). In vitro antimicrobial activity of Boswellia sacra frankincense oils grown in different locations in the Dhofar region (Oman). Antibiotics, 9(4), 195.
Al-Harrasi, A. and Al-Saidi, S. (2008). Phytochemical analysis of Boswellia sacra (Omani Luban) botanically certified oleogum resin essential oil. Molecules, 13(9), 2181-2189.
Response 2: The study was justified more clearly by including the mentioned references and adding the scientific value and the novel things in the present paper by adding this information to the end of introduction section:
“Thus, the present study screened the antimicrobial activity of frankincense oil using the well diffusion method, and determined its MICs, MBCs, and MFCs. This is in contrast to some previous studies that reported the antimicrobial activity of the volatile constituents of essential oils [24] and the antimicrobial activity of frankincense hydro-distilled frankincense essential oil by determination of the MIC only [13]. This study also allowed better characterization of the antimicrobial activity of frankincense oil by using broth micro-dilution method coupled with optical density readings. Moreover, to our knowledge, this is the first study that developed a method to test the antimicrobial activity of frankincense smoke against airborne microbes. This study analyzed the chemical profile of the steamed-distilled frankincense oil in comparison to the previous studies that focused more on the determination of the phytochemical profile of frankincense hydro-distilled oil and showed the presence of various monoterpenes and sesquiterpene [25].
Point 3: Material and methods. 2.1. Include identification voucher.
Response 3: A picture of the samples used in the study was added; Supplementary Figure 1 in the supplementary file as no identification voucher was deposited. Authors personally got the samples from the trusted trader.
Point 4: 2.6. The method is validated. Are there any other studies that support the effectiveness of this method?
Response 4: We included a reference method for collecting airborne microbes. We could not find studies that test the antimicrobial activity of smoke against pure cultures of microbes as it is described in our study. We found one study that used unidentified herbal smoke in Kenya to test the efficiency of disinfecting camel containers but it did not use pure microbial cultures (Wanjala, N.W., Matofari, J.W. & Nduko, J.M. Antimicrobial effect of smoking milk handling containers’ inner surfaces as a preservation method in pastoral systems in Kenya. Pastoralism 6, 17 (2016). https://doi.org/10.1186/s13570-016-0064-y).
Point 5: Results
How does this study differ from the other reported studies? What is new information reported?
Chemical characterization. The results of the characterization are consistent with the literature. In this sense, consider that originality can be improved. Highlight if novel results were found.
Response 5: The novel things in this study have been highlighted in the introduction clearly and then the novel results have been highlighted especially the part of the antimicrobial activity of the smoke. The main focus of the study is the microbial part (antimicrobial activity of oil and smoke). The chemical analysis was included as it is explained in the introduction to check the constituents of oils of both grades. The comparison with other studies is explained in the discussion. The antimicrobial activity of oil reported the results of inhibition zones using well diffusion method. Others included mostly MIC results. Also, we could not find in literature MBC/MFC; the cidal concentrations. The smoke results are also reported here for the first time according to our knowledge. Optical density results allowed us to show the pattern of inhibition of microbial growth using frankincense oil (Figure 4). Also we did not find any previous study that provides a detailed description as this for the antimicrobial activity of frankincense oil and smoke as this study.
I added this explanation in introduction to make it clear the difference and the novel things in this study as follows: “Thus, this study screened the antimicrobial activity of frankincense oil using the well diffusion method, and determined its MICs, MBCs, and MFCs. This is in contrast to some previous studies that reported the antimicrobial activity of the volatile constituents of essential oils [24] and the antimicrobial activity of frankincense hydro-distilled frankincense essential oil by determination of the MIC only [13]. This study also allowed better characterization of the antimicrobial activity of frankincense oil by using broth micro-dilution method coupled with optical density readings. Moreover, to our knowledge, this is the first study that developed a method to test the antimicrobial activity of frankincense smoke against airborne microbes. This study analyzed the chemical profile of the steamed-distilled frankincense oil in comparison to the previous studies that focused more on the determination of the phytochemical profile of frankincense hydro-distilled oil [25].”
In results:
“In the literature, there is a lack of information regarding bactericidal and fungicidal concentrations of frankincense oil.”
“Average results of analysis of frankincense smoke solid particles are presented in Figure 5 for the first time.”
Point 6: Antimicrobial activity
Report the results as µg/mL, to perform comparative analyses. Same previous comment, highlight if novel results were found.
Response 6: Results were reported in µg/mL as well in section 3.3, Table 1, and table 2. Novel results were reported as in the previous response.
Point 7: 3.2. First, describe the results for bacteria and later yeast and fungi.
Response 7: It was done as follows:
” All bacteria, yeasts and the spores of molds were susceptible to Hojari and Sha’bi frankincense oils (Table S2, Figure 1). For bacteria, the smallest inhibition zone was produced with E. coli using the Sha’bi and the Hojari oils (mean= 10.4 ± 1.3 and 10.4 ± 0.9 mm respectively). The inhibition zones that were produced by the oils on S. aureus and Bacillus spp. were slightly larger than those produced on E. coli and P. aeruginosa. For fungi, the largest growth inhibition zone was produced with S. cerevisiae using both Sha’bi and Hojari oils (mean diameter= 37.8 ± 6.7 and 36.3 ± 8.2 mm respectively) followed by F. solani with Sha’bi and Hojari oils; mean= 29.1 ± 4.1 mm and 28.2 ± 2.4 mm, respectively (Figure 3). The smallest inhibition zone was produced with A. flavous using Hojari oil (mean= 10.4 ± 1.3 mm). A. flavous followed by P. citrinum were the most resistant molds to Sha’bi and Hojari oils. The yeast S. cerevisiae was slightly more susceptible than C. albicans. Regarding molds, the greatest inhibition zone was produced with F. solani.”
Point 8: 3.3. Describes more precisely the most relevant results. The most active treatments.
Response 8: The relevant results were described more clearly; “E. coli was the most susceptible bacterium and showed MBC of 3.13% or 1.56% as compared to MBC of 12.5% with all other bacteria.”
Statistical results explained better if there were significant differences between treatments as follows: “Only the type of microorganism was found to influence the MIC of oils significantly (ANOVA, P= 0.0335, α= 0.05). The MIC was not significantly affected by the grade of frankincense or the method of determination (ANOVA, P= 0.3156 and 0.1085 respectively, α= 0.05). No significant interaction on the MIC values was detected between the method and the grade (ANOVA, P= 0.3739, α= 0.05), the method and the type of microorganism (ANOVA, P= 0.8523, α= 0.05) or the grade and the type of microorganism (ANOVA, P= 0.5995, α= 0.05).”
Point 9: 3.5. To which bacterium, fungus, or yeast does the sample b14, b17, b18, m7 and m1 correspond ? Because they did not carry out biochemical or molecular tests.
Response 9: Although it would be interesting and informative, the airborne isolates were not further analyzed in the experiment that was done to check the antimicrobial activity of frankincense smoke against airborne microbes. This point was reported clearly in section 2.6.1 as follows:
“These organisms were identified as three Gram-positive bacteria (B14, B17, and B18), one mold (M1), and one yeast (M7) based on their morphology and microscopic characteristics with no further biochemical or genetic identification done for the airborne isolates.”
Point 10: Discussion
Line 561-562-The results in the table show that E. coli was more susceptible. Is there a typing error? It does not agree with what is discussed in the table.
Response 10: No typing error. The sentence refers to Figure 2, the results of the well diffusion method and not the table. In this part we compared the results of inhibition zone diameter techniques (well diffusion and disc diffusion) and not the broth dilution method for determination of MIC. The word “Figure 2” was added for better understanding as follows: “Utilizing the well technique in this study (Figure 2), S. aureus and Bacillus spp. were slightly more susceptible to Sha’bi and Hojari frankincense oil than E. coli and P. aeruginosa.”
Regarding E. coli (Table 2) result in broth dilution method (MIC), differences between results of agar diffusion and broth dilution methods with comparison with previous study was explained as follows in the discussion “Other researchers [45] considered estimation of bactericidal concentrations of essential oils to be more precise than the agar well technique. They found inhibition zones produced by rosemary essential oil against S. aureus and Campylobacter jejuni to be 5.9 and 9.3 mm, respectively. However, their bactericidal concentrations were 0.1% and >1% for S. aureus and C. jejuni, respectively.”
Point 11: Line 571-577. P. aeruginosa and E. coli are Gram -, explain why one is more susceptible than the other. Are the antimicrobial found results important? Can they be classified as extracts with high antimicrobial potential? Compare with natural product classifications.
Response 11: This has been explained in discussion and also with comparison to our previous study as follows:
“Species variation in each bacterial group of Gram-positive and Gram-negative bacteria may lead to variation in their susceptibility to antimicrobial agents. For instance, in this study, E. coli had lower MIC than P. aeruginosa. Likewise, our previous study using the same strains of E. coli and P. aeruginosa found lower MIC results for E. coli (0.78%) than P. aeruginosa (1.56%) with both clove and thyme essential oils [27].”
Regarding the importance of antimicrobial activity results, this has been highlighted in different sections and also according to the other reviewer, The possible effect of frankincense on human and food microbiota have been added as follows:
“Thus, frankincense oil used in this study can be considered to have moderate antimicrobial activity. High MIC values might indicate that the active components are present in the extract at low concentrations as a result of the extraction method [5]. This supposes that, detecting the antimicrobial activity of the crude extracts is essential regardless of their potency.”
“Frankincense is added to food for its health benefits, however, it is important to determine its effect on gut microbiota and probiotics intended to deliver live cultures to the gut, taking into consideration different factors such as the concentration of frankincense and the type of its formulation.”
Point 12: Describe the possible mechanism of action of the treatments evaluated based on the main compounds.
Response 12: This has been explained in the discussion as follows:
“In this study, α-pinene was a major chemical compound found in all oil samples and it might be the principle active chemical and thus should be investigated further to determine its antimicrobial potential as a pure compound and its mechanism of action. Essential oils are known to interact with cell membranes and this can affect the function of different membrane components such as receptors, enzymes, transport systems, and ion channels. It was reported that synergy between monoterpenes like pinene and sesquiterpenes such as β-caryophyllene can occur. The compounds α-humulene and myrcene (which were present in the oil used in this study) had good antimicrobial activity against some Gram-positive and Gram-negative bacteria and some yeasts [50]. Thus, the essential oil used in this study contained chemical compounds that are known to have activities against bacteria, yeasts and molds which might explain its broad-spectrum antimicrobial activity. However, purification and identification of specific bioactive chemical compounds in Hojari and Sha’bi frankincense oils require further investigation.”

Reviewer 2 Report
Article foods-2610774 investigates the antimicrobial activity of frankincense oil and smoke against certain microorganisms. Despite the fact that frankincense research has been going on for a long time, this work makes a significant contribution to the overall scientific picture. The methodology is described and executed very carefully and the results are beyond doubt. The discussion was comprehensive. The work was done at a high level. The work corresponds to the subject of the journal, the declared section and the special issue.
Small questions and comments:
1. In point 2.6.2 the exposure to smoke lasted 2 hours. This is quite a long time. How was the duration of smoke exposure chosen?
2. The fact that the plastic substrate dissolves with frankincense oil is very interesting. What substances, according to the authors, are responsible for this? Can this fact give impetus to the directed destruction and recycling of plastics?
3. You found no significant difference between premium and inferior frankincense in terms of antimicrobial activity. You make the assumption that this is due to the freshness of the extracted fractions. Then it would be interesting to confirm this hypothesis in the future. Conduct research on the dependence of antimicrobial activity on the duration of storage of oil of the highest and lowest grades. In the introduction, you write very interestingly how you bought incense samples based on your long experience as an incense merchant.
4. Adding frankincense to food can have negative effects on the human microbiota?
5. Line 689. The addition of frankincense to yogurt may cause a decrease in the titer of lactic acid bacteria. So what is the purpose of this supplement?
Author Response
Point 1: In point 2.6.2 the exposure to smoke lasted 2 hours. This is quite a long time. How was the duration of smoke exposure chosen?
Response 1: same duration for collecting airborne microbes was used. Also, from our experience of how fumigation is usually done. It was not easy to test different times of microbial exposure to smoke as we had also to work with smoke detectors in the building.
This was clarified in the article as follows:
“Once the burning was complete and the smoke had been generated, the test plates were opened and left exposed to the smoke for two hours to make it possible for the active components in the smoke to contact the test microbes”.
Point 2: The fact that the plastic substrate dissolves with frankincense oil is very interesting. What substances, according to the authors, are responsible for this? Can this fact give impetus to the directed destruction and recycling of plastics?
Response 2: For explanation of this in this article we added this information:
“Some compounds such as cymene, terpinene, limonene, and phellandrene which are frequently found in essential oils including frankincense oil were previously reported to be good solvents for polystyrene foams [33]. It will be interesting to identify which components in frankincense oil responsible for dissolving different types of plastics and determine their applicability. “
Point 3: You found no significant difference between premium and inferior frankincense in terms of antimicrobial activity. You make the assumption that this is due to the freshness of the extracted fractions. Then it would be interesting to confirm this hypothesis in the future. Conduct research on the dependence of antimicrobial activity on the duration of storage of oil of the highest and lowest grades. In the introduction, you write very interestingly how you bought incense samples based on your long experience as an incense merchant.
Response 3: We will include this in our future research, usually essential oils have specific shelf life after which, their activity may decrease.
Point 4: Adding frankincense to food can have negative effects on the human microbiota?
Response 4: In industry, frankincense is added to food for its various health benefits and this attracts consumers. Recently, many frankincense supplements have been produced as well, for their positive effects on health (Examples anti-inflammatory and as preventive measures for diseases that are mentioned in the manuscript).
I added this information for clarification:
“Frankincense is added to food for its health benefits, however, it is important to determine its effect on gut microbiota and probiotics intended to deliver live cultures to the gut, taking into consideration different factors such as the concentration of frankincense and the type of its formulation.”
Point 5: Line 689. The addition of frankincense to yogurt may cause a decrease in the titer of lactic acid bacteria. So what is the purpose of this supplement?
Response 5: I combined the answer to this point with the previous point. “Frankincense is added to food for its health benefits, however, it is important to determine its effect on gut microbiota and probiotics intended to deliver live cultures to the gut, taking into consideration different factors such as the concentration of frankincense and the type of its formulation.”
